# Evaluation of the Anticoccidial Activity of Sheep Bile against *Eimeria stiedae* Oocysts and Sporozoites of Rabbits: An In Vitro Study

**DOI:** 10.3390/vetsci9120658

**Published:** 2022-11-25

**Authors:** Mutee Murshed, Saleh Al-Quraishy, Mahmood A. Qasem

**Affiliations:** Department of Zoology, College of Science, King Saud University, P.O. Box 2455, Riyadh 11451, Saudi Arabia

**Keywords:** *Eimeria stiedae*, sheep bile, oocysts, sporozoites, inhibition, in vitro

## Abstract

**Simple Summary:**

Coccidiosis is a common disease that infects rabbits, causing substantial problems, such as severe diarrhea, feces containing mucus or blood, weight loss, reduced reproductivity, and a high death rate. The widespread use of the currently available anticoccidial drugs has led to widespread drug resistance, prompting the search for alternative drugs. The bile of animals, such as poultry, cattle, sheep, etc., is a promising candidate for alternative bile-related therapeutic strategies. This study highlights the potential of sheep bile as an anticoccidial agent by demonstrating its capacity to inhibit the sporulation of coccidian oocysts, as well as its anti-sporozoite activity.

**Abstract:**

Coccidiosis is one of the most common infectious diseases that causes digestive problems in rabbits, leading to global economic losses. This study was conducted to evaluate the effects of bile obtained from sheep gallbladder on the sporulation and morphology of *Eimeria stiedae* oocysts and sporozoites affecting rabbit liver cells and to determine the best concentration for sporulation inhibition. Sporulation inhibition per milliliter was measured in samples exposed to five concentrations of sheep bile (SB) in a 2.5% potassium dichromate solution: 12.5%, 25%, 50%, 75%, and 100% concentrations for oocysticidal activity and 125, 250, 500, 750, and 1000 μg/mL concentrations for antisporozoidal activity. A bioassay was performed to assess the in vitro anticoccidial activity of sheep bile against *E. stiedae* oocysts and sporozoite sporulation. In this assay, six-well plates with 5 mL of bile containing 1000 oocysts showed unsporulated oocysticidal activity after 48, 72, and 96 h and antisporozoidal activity after 12 and 24 h. A chemical assay was performed via infrared spectroscopy to investigate the presence of several anticipated active chemical compounds in sheep bile. Sheep bile was able to inhibit *E. stiedae* oocysts at 100% and 75% concentrations by about 91% and 81%, respectively. In addition, SB had the highest inhibition of *E. stiedae* sporozoite viability (92%) at a concentration of 1000 μg/mL and had the lowest inhibition of 8% at a concentration of 125 μg/mL. An increase in the incubation time and a higher dose generally increased the inhibition rate. The results showed that sheep gallbladder bile is effective due to its inhibitory potential and effect on the coccidian oocyst sporulation of *E. stiedae*. Further studies are needed to determine the precise active chemicals present in SB and their modes of action and application in vivo.

## 1. Introduction

*Eimeria stiedae* causes hepatic coccidiosis in rabbits. For more than a century, coccidiosis has been established as one of the most frequent parasitic diseases in rabbit colonies, causing severe adverse outcomes. Its pathogenicity ranges from mild to severe, including weight loss, mild and intermittent-to-severe diarrhea, with mucus or blood in feces, resulting in dehydration and decreased rabbit breeding rates [1,2]. Adult rabbits who carry a coccidial infection and are typically asymptomatic can cause infections in young populations, with morbidity and mortality rates reaching 90% and 60%, respectively [3,4,5]. *E. stiedae* parasitizes epithelial cells in the bile duct of rabbits and is particularly pathogenic among intracellular extraintestinal coccidia species. It is one of fourteen Eimeria species infecting rabbits worldwide [6,7]. In recent years, rabbit farming has been increasingly commercialized and considered a major source of income. In addition to providing a popular source of protein among consumers because of its low cholesterol and fat content, rabbit meat contains high quantities of vital amino acids, making rabbit meat one of the world's most valuable animal resources [6]. However, coccidiosis remains one of the most common gastrointestinal infectious diseases in rabbits [8]. According to Chapman, coccidiosis costs the US rabbit industry USD 127 million per year, and similar losses may occur worldwide [9]. The increased resistance to anticoccidial treatment has induced a search for alternatives [10,11]. For over two thousand years, 44 different animal bile solutions, including human gallbladder fluids, have been used as pharmaceuticals in traditional Chinese medicine (TCM) to treat a variety of diseases, which have been authenticated by Chinese medicinal researchers [4] because of their membranolytic and detergent characteristics [12,13]. Sheep bile is a complex fluid that passes through the biliary system and into the small intestine, comprising water, salts, and organic molecules, such as bile acids, cholesterol, phospholipids, and bilirubin. Bile acids play important roles due to their antimicrobial characteristics [14,15], which are important for the inhibition of bacterial activity in the proximal small intestine [14,16]. Due to their antibacterial activity and anticoccidial chemotherapeutic effects, bile acids are currently sold as therapeutic alternatives after a series of research trials investigating their efficacy. They are also a less expensive method to eliminate coccidiosis. In traditional Chinese medicine (TCM), gallbladder fluids and some bile ingredients derived from other animals, as well as plants, medications, and other sources, are used to treat infectious and non-infectious diseases, including both chronic and acute diseases such as malaria [17]. The major components present in bile, such as specific bile salts, bile pigment bilirubin, and glucuronides, as well as minor bile components, such as vitamins A, D, E, and K, and melatonin synthetic compounds, can be found in medicinal plants and the bile derived from ruminants, such as chickens, cattle, sheep, etc. Therefore, they may be promising candidates for similar, alternative therapeutic purposes of bile based on chemical composition and pharmacological actions [18]. Animal bile is shown to improve liver function, dissolve gallstones, suppress bacterial and viral proliferation, and have anti-inflammatory, antioxidant, anodyne, anticonvulsive, antiallergic, anticongestive, antidiabetic, and antispasmodic properties [4,18,19]. Bile is rich in mineral ions and trace amounts of proteins, including mucin glycoproteins [20], and has high levels of antioxidants, such as bilirubin, glutathione, vitamin E, and melatonin (N-acetyl-5-methoxytryptamine) [21]. Recently, in modern therapeutic research, bear bile has been shown to contain a wide range of pharmacological properties, including hepatoprotective, antibacterial, antiviral, anti-inflammatory, antigallstone, and hypolipidemic activities, etc. [22,23]. Goat bile has been therapeutically used in TCM to facilitate the treatment of optic atrophy, acute hemorrhagic conjunctivitis, and various infectious skin illnesses [24]. The primary components of chicken bile acid are taurocholate (TC), taurochenodeoxycholic (TCDC), and tauroglycocholate (TAC) [25]. Investigations are underway on whether carcass residue (bile) can be utilized to treat coccidiosis in birds [26,27]. Toltrazuril is an effective drug in preventing coccidiosis in rabbits, pigs, sheep, and cattle [28]. The first clinical trial of 2.5% toltrazuril for the treatment of rabbit intestinal coccidiosis showed that 25 ppm toltrazuril dissolved in water for two days was effective in treating the disease [29,30]. Taking these into consideration, the purpose of this study was to assess the in vitro potential of healthy sheep bile against the sporulation and morphology of *Eimeria species*, namely *E. stiedae* oocysts. 

## 2. Materials and Methods

### 2.1. Preparation of Bile 

Bile was obtained from the intact gallbladders of local sheep in the Riyadh Automated Slaughterhouse (Saudi Arabia). Gallbladders were isolated from 10 healthy sheep and sterilized with 70% alcohol, before being extracted using a syringe, transferred to a clean tube, and stored at 4 °C until use. 

### 2.2. Infrared Spectroscopy

A small amount (1 mL) of sheep bile was combined with a high amount of potassium bromide powder (1:99 wt %) and processed to obtain uniform consistency; then, the mixture was coarsely crushed and deposited in a pellet-forming die. An optical spectrometer (NICOLET 6700 from Thermo Scientific) was used to conduct Fourier transform–infrared spectroscopy (FT-IR). The number of waves reflected the maximum absorption (cm^−1^). Spectra were recorded at 25 °C with a resolution of 4 cm, ranging from 4000 cm^1^ to 400 cm^1^. 

### 2.3. Isolation and Preparation of Eimeria stiedae 

Oocysts of the parasite *E. stiedae* were isolated from feces, gallbladders lumen, and necrotic hepatic lesions of rabbits that had spontaneously contracted the infection, and a surface was sterilized with sodium hypochlorite and then rinsed four times in sterile saline, as previously described [31]. Flotation was used to concentrate and extract oocysts using a saturated NaCl solution [32]. Oocyst sporulation was carried out in a moist chamber at temperatures ranging from 24 to 26 °C and humidity of approximately 60% for one week. The sporulated oocysts were kept in an aqueous solution of 2.5% potassium dichromate at 4 °C until they were used in the experiments. In addition, the isolated *E. stiedae* parasites were kept alive in our parasitology laboratory by regularly injecting them into young rabbits. According to Levine, the oocyst count (oocysts per gram of OPG) was calculated using the McMaster technique [33].

### 2.4. In Vitro Effect of Sheep Bile on Inhibition of Sporulated Oocysts 

An in vitro sporulation inhibition assay was performed to investigate the effects of sheep bile on the oocyst sporulation of *E. stiedae*, and its disinfectant activities were evaluated using 6-well plates. Each well contained 5 mL of bile suspension at each concentration, which was diluted with distilled water, inoculated with an equal number of unsporulated oocysts, and incubated. Then, 430 mL of 1 × 10^4^ unsporulated oocysts were added to a 2.5% potassium dichromate solution (K_2_Cr_2_O_7_) and distilled water and exposed to five concentrations of sheep bile (V/V; 12.5%, 25%, 50%, 75%, and 100%), with a potassium dichromate solution (PD) as a control group. The 6-well plates were partially covered to allow oxygen flow, and incubation was carried out in an incubator at 25–29 °C with water in two Petri dishes to maintain humidity at 60–80%. The plate contents were stirred [34], and each dilution was microscopically examined at 24, 48, 72, and 96 h. Sporocysts were examined under an inverted microscope at 40× magnification to confirm oocyst sporulation. The number of oocysts left in a total of 50,000 oocysts was counted to estimate the population of dead oocysts [10]. Three replicates were performed for each concentration, and the experiment was then repeated to corroborate the findings. Oocysts with four sporocysts were defined as sporulated oocysts, and oocysts with abnormal sporocysts (in size and shape) were counted. A McMaster chamber was used for the measurement of sporulated and unsporulated oocysts. The percentages of sporulation and inhibition were estimated using the following equation: sporulation (%) = number of sporulated oocysts/total number of oocysts × 100. As defined by [35], sporulation inhibition (SI%) = sporulation (SP%) of control sporulation (SP%) of treated/sporulation of control [10]. 

### 2.5. In Vitro Anti-Sporozoite Effect of Sheep Bile 

The oocyst samples stored in K_2_Cr_2_O_7_ were washed several times with phosphate-buffered saline (PBS) (pH 7.4). We centrifuged 15 ml tubes (Falcon) at 1008× *g* several times for 10 min until K_2_Cr_2_O_7_ was eliminated. The oocysts were incubated in a water bath at 41 °C and shaken for 60 min. The suspension was centrifuged at 1008× *g* for 10 min, resuspended in PBS, and rinsed once with PBS. The sporozoites were counted using a McMaster chamber. Twenty-four wells (four 6-well plates) were used to evaluate sporozoite activity in vitro. Briefly, 1 mL of the test solution was mixed and added to 1 mL of the parasite suspension containing 1000 sporozoites to yield a total volume of 2 mL of each concentration of sheep bile (125, 250, 500, 750, and 1000 μg/mL). For comparison, toltrazuril was used as the reference medication, K_2_Cr_2_O_7_ was used as a negative control, and 50 µL/mL toltrazuril was used as a positive control. The results were evaluated after 12 and 24 h, and the experiment was repeated 3 times for each concentration and control treatment under identical settings. The number of viable and nonviable sporozoites was calculated, and the percentage of viability was estimated by counting the number of viable sporozoites in a total of 100 sporozoites. The percentage inhibition of viability was calculated as follows: Inhibition of viability (Vi) = (Vi% of control − Vi% of bile/Vi% of control × 100).

### 2.6. Statistical Analysis

Sigma Plot software was used for the statistical analysis (version 11, Systat software, Inc., San Jose, CA, US). The total impact of each therapy was determined using a one-way ANOVA, and the treatments were individually compared using Duncan’s test. Values are expressed as mean ± SD, and *p* ≤ 0.05 was considered statistically significant.

## 3. Results

### 3.1. Spectroscopy Results 

The results of infrared spectroscopy reveal several active chemical compounds in sheep bile with major bands at 3425.53 cm^−1^, 2093.10 cm^−1^, 1641.41 cm^−1^, 1045.64 cm^−1^, and 410.42 cm^−1^, respectively (Figure 1). However, other compounds were also detected (Table 1).

### 3.2. Oocysticidal Activities of Sheep Bile In Vitro 

The oocysticidal activity of different concentrations of sheep bile against the isolated *E. stiedae* was investigated in vitro. The highest efficacy among the tested sheep bile concentrations was recorded after 96 h of exposure to a concentration of 100% of sheep bile, leading to approximately 91% inhibition of sporulation. In comparison, the control group (K_2_Cr_2_O_7_) revealed a significantly high level of oocyst sporulation. Conversely, a low concentration of 12.5% of BS reduced the rate of oocysticidal inhibition. Other SB concentrations (75%, 50%, and 25 mg/mL) showed unequal efficacy depending on the tested concentration and incubation time. 

After 48 h of incubation with SB at concentrations of 75% and 100%, the unsporulated *E. stiedae* oocysts exhibited no sporulation. In contrast, the oocysts incubated with 2.5% potassium dichromate and SB with concentrations of 12.5%, 25%, and 50% showed varying degrees of sporulation (Figure 2).

Unsporulated *E. stiedae* oocysts were inhibited after incubation with SB at concentrations of 75% and 100% for 72 h, whereas the number of sporulated oocysts continued to increase with an increase in incubation time with the 2.5% potassium dichromate solution, and 12.5%, 25%, and 50% SB concentrations showed different levels of sporulation (Figure 3).

After 96 h of incubation with 75% and 100% concentrations of SB, unsporulated *E. stiedae* oocysts showed no sporulation. However, the number of sporulated oocysts continued to increase with the increased duration of incubation with 2.5% potassium dichromate, and SB exhibited different levels of sporulation at 12.5%, 25%, and 50% concentrations (Figure 4).

The results of this study reveal the in vitro effects of the duration of sporulation and different bile concentrations on the sporulation (%) and non-sporulation (%) rates of *E. stiedae* oocysts. The percentage of sporulation increased with an increase in incubation time, whereas the percentage of non-sporulation decreased. The rate of sporulation inhibition significantly increased with an increase in incubation time up to 72 h (*p* ≤ 0.05); however, the rate of sporulation inhibition did not significantly differ between 72- and 96-h exposure (Figure 5).

Overall, the experimental groups significantly affected the rates of sporulation (%) and non-sporulation (%). SB concentrations at 100%, 75%, and 50% had the highest non-sporulation rates and lowest sporulation rates (*p* ≤ 0.05). Concentrations of 25% and 12.5% SB had higher rates of non-sporulation and inhibition of sporulation than the control group (*p* ≤ 0.05). 

### 3.3. Antisporozoidal Activity of Sheep Bile In Vitro 

Different concentrations of sheep bile showed a concentration-dependent inhibition of the coccidial viability of *E. stiedae* sporozoites. The results show the effect of sheep bile on sporozoite inhibition, incubation time, and bile liquid type in a concentration-dependent manner. Antisporozoidal activity was observed at the high concentration of 1000 µg/mL, and inhibition rates significantly increased when concentration was increased, thus improving the bile efficacy. At 750 and 1000 µg/mL concentrations of bile, the highest inhibitory effects against *E. stiedae* were observed at 12 and 24 h compared with the control groups, control-I (K_2_Cr_2_O_7_) and control-II (toltrazuril). In contrast, 500, 250, and 125 µg/mL concentrations revealed less inhibitory activity. Similarly, Figure 6 shows that the inhibition rate increased with an increase in incubation time. The high bile concentration of 1000 µg/mL reduced *E. stiedae* viability by 92%. The lowest efficacy was 8% at a concentration of 125 μg mL after 24 h incubation against *E. stiedae*. A high SB concentration with increased incubation time revealed limited inhibition of *E. stiedae* viability, with 0.95% at 125 μg/mL of SB. As the concentration of SB decreased, the percentage of viability inhibition decreased accordingly (Figure 6).

## 4. Discussion

Rabbit coccidiosis is caused by *Eimeria* spp., a pathogenic species that inhabits the liver and intestines. *E. stiedae* was identified in fecal samples containing oocysts. In this study, the effect of healthy sheep bile on the oocyst sporulation of *E. stiedae* was tested in vitro at several concentrations (SB: 12.5%, 25%, 50%, 75%, and 100%) by measuring over different time periods (48, 72, and 96 h). The setup was tested for anti-sporozoite activity at 125, 250, 500, 750, and 1000 µl/ml after 12 and 24 h. The current study demonstrated the inhibitory effects of sheep bile on coccidian oocyst sporulation. According to our results, sheep bile concentrations of 100% and 75% exhibited oocyst activity against *Eimeria species*. These concentrations were found to be highly effective in the sporulation inhibition of *E. stiedae* oocysts. Studies on sheep bile compounds are limited. To the best of our knowledge, this is the first study in Saudi Arabia on the use of animal bile to prevent coccidiosis oocyst sporulation in *E. stiedae*. Our results correspond with those of Tremblay et al., who showed that the bile acid in the ileum is an antibacterial agent that may be employed as a treatment option for intestinal bacterial infections [24]. Similarly, animal bile and commercially available bile acids have been studied in experimental allergic disease models for their antiallergic properties. In mouse models used for the investigation of delayed hypersensitivity and picryl-chloride-induced contact dermatitis, pig bile exerted a significant protective effect. In a previous study, PC-CD was inhibited by using the herb Fleurs (dried bear gallbladder) [36]. Similar to our results, Cedric et al. observed in vitro sporulation inhibition using Psidium guajava leaf extracts in four *Eimeria species* [37]. Since condensed grape tannins have been shown to block endogenous enzyme activity (such as mannitol dehydrogenase, mannitol-1 phosphatase, mannitol-1 phosphate dehydrogenase, and hexokinase) [37], sheep bile (which contains tannins) may reduce the rate of sporulation by inhibiting or inactivating the enzymes responsible for sporulation, similar to helminth eggs [38]. 

Jones et al. suggested that bile extracts might enter oocyst cell walls and induce a loss of intracellular components [39]. Thus, in the present study, sheep bile may have entered the oocysts’ walls and damaged the cytoplasm (sporont), as demonstrated by the appearance of abnormal sporocysts in the oocysts exposed to higher concentrations [38]. 

The inability of potassium dichromate to limit sporulation could be explained by the fact that it is also a bactericidal drug; therefore, it boosts oocyst sporulation. Potassium dichromate destroyed bacteria in a sample containing coccidian oocysts, encouraging coccidian oocyst sporulation [40]. Thus, bacteria, if present, may interfere with oocyst sporulation, presumably by competing for nutrients or feeding on oocysts.

These results also suggest the potential therapeutic role of bile acids in the treatment of enteric bacterial infections [24,41], for which alcohol sulfates and N-acyl amidites in C27 and/or C24 bile acids can be used. Bile acids are a molecular combination of congeners formed from cholesterol in the liver. Hundreds of species are currently known that naturally generate bile acids, with humans being the most developed [23,42].

Bile is used to dislodge intestinal worms from dogs, and it plays a critical role in treating infantile undernourishment caused by gastrointestinal disruptions and trematodes. According to paleopathological findings, these infestations are most likely caused by roundworms (nematodes) [19,43,44]. 

Modern research on the physiological and physicochemical properties, nuclear receptor regulation, and the homeostasis of bile acids have shed light on the possible pharmacological mechanisms involved in different animal bile compounds. These findings confirm the success of millennia-old heuristic therapies practiced in TCM [45,46]. Our findings are consistent with those of Remmal, who independently evaluated carvacrol, isopulegol, thymol, eugenol, and carvone, the primary components of essential oils, discovering an oocysticidal efficacy against coccidiosis [26]. Furthermore, over the last two decades, bile acids have been suggested as potent regulatory agents in the gastrointestinal tract and liver due to the presence of G-coupled protein receptors, activation of specific nuclear receptors, and multiple cell signaling pathways [41,46]. 

Sporozoite viability in control-I (K_2_Cr_2_O_7_) and control-II (PBS) was also investigated in previous studies [40,47]. Our results support the results of other studies regarding the inhibitory effect of Curcuma longa on *E. tenella* sporozoite activity and the secretion of chicken bile on *E. stiedae* activity [48,49]. Schubert et al. found that extracellular calcium and Ca^2+^ signaling are required for the penetration of *E. tenella* sporozoites into host cells [50]. Bile extracts have been demonstrated to activate and desensitize calcium channels [51]. This is consistent with the study by Cedric et al., who revealed the potential contribution of P. macrophylla extracts to the observed inhibition of sporozoite viability by affecting calcium-mediated signaling in the sporozoites [37]. 

## 5. Conclusions

Our findings provide evidence for the use of sheep bile as an anticoccidial agent by demonstrating its inhibitory capacity against the sporulation of coccidian oocysts and anti-sporozoite activities. Further studies are needed to confirm the protective effects of sheep bile against hepatic infections caused by *E. stiedae* in vivo and to identify safe, active compounds that can be produced from cheap natural products. 

## Figures and Tables

**Figure 1 vetsci-09-00658-f001:**
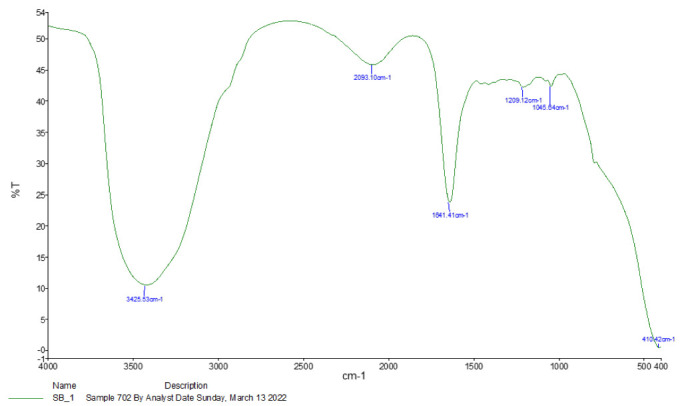
Infrared spectroscopy results of sheep bile samples. The results were obtained using a Nicolet 6700 FT-IR spectrometer in the range of 400–4000/cm^−1^.

**Figure 2 vetsci-09-00658-f002:**
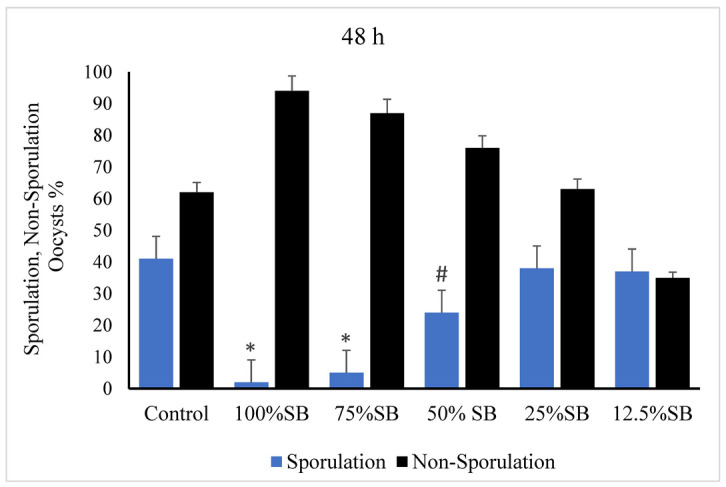
Effect of sheep bile on sporulation of *E. stiedae* oocysts at 48 h in vitro. Significance (*): *p*-value ≤ 0.001 and (#): *p*-value ≤ 0.05. (SB): sheep bile. (h): hours.

**Figure 3 vetsci-09-00658-f003:**
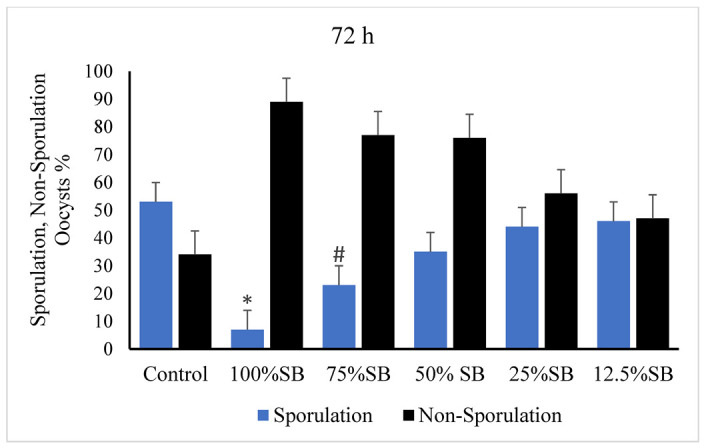
Effect of sheep bile on the sporulation of *E. stiedae* oocysts at 72 h in vitro. Significance (*): *p*-value ≤ 0.001 and: (#) *p*-value ≤ 0.05. (SB): sheep bile. (h): hours.

**Figure 4 vetsci-09-00658-f004:**
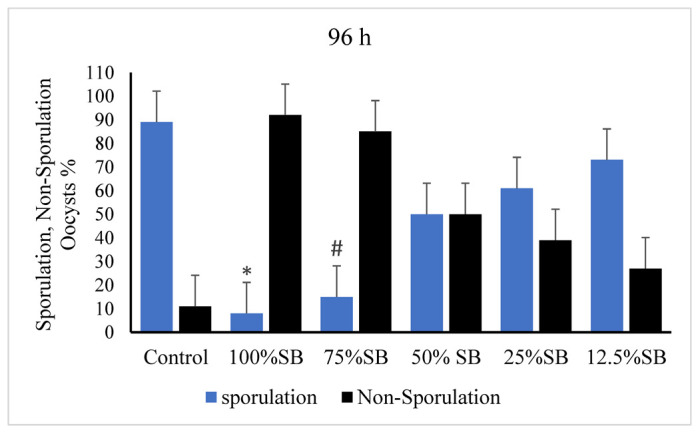
Effect of sheep bile on sporulation of *E. stiedae* oocysts at 96 h in vitro. Significance (*): *p*-value ≤ 0.001 and (#): *p*-value ≤ 0.05. (SB): sheep bile. (h): hours.

**Figure 5 vetsci-09-00658-f005:**
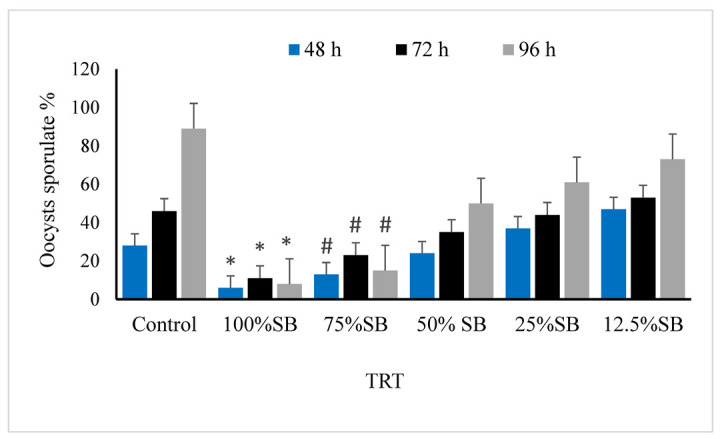
Main effects of sheep bile on sporulation% and non-sporulation% of *E. stiedae* oocysts at different concentrations with varying contact time and treatment effects at 48, 72, and 96 h in vitro. The statistical significance was compared with 2.5% potassium dichromate as control. (*): indicates *p*-value ≤ 0.001 and (#): indicates *p*-value ≤ 0.05. (SB): sheep bile. (h): hours.

**Figure 6 vetsci-09-00658-f006:**
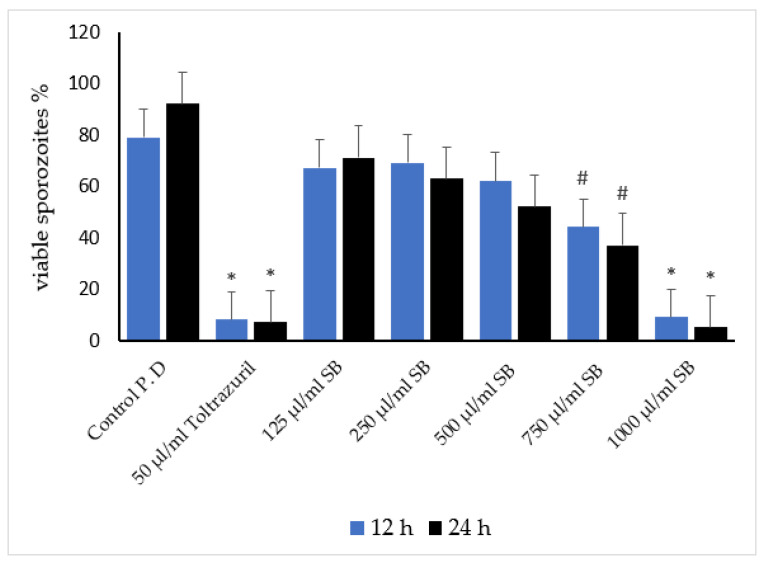
In vitro effects of sheep bile on inhibition of *E. stiedae* sporozoite viability at various concentrations after 12 and 24 h. The significance was compared with 2.5% potassium dichromate as a control and 50 µL/mL of toltrazuril. (*) indicates a *p*-value of ≤ 0.001, while (#) indicates a *p*-value of ≤ 0.05. (SB): sheep bile. (P. D): potassium dichromate. (h): hours.

**Table 1 vetsci-09-00658-t001:** FT-IR spectrum of sheep bile based on frequency range.

Absorption (cm^−1^)	Appearance	Transmittance (%)	Group	Compound Class
3425.53	Medium	12	N-H stretching	Aliphatic primary amine
2093.10	Strong	47	N = C = S stretchy	Isothiocyanate
1641.41	Strong	25	C = C stretching	Alkene
1209.12	Strong	37	C-O stretching tertiary	Alcohol
1045.64	Strong and broad	35	CO-O-CO stretching	Anhydride
410.42	Strong	3	C-H bending	1,2-disubstituted

## Data Availability

Not applicable.

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
