# Peer review of "Evaluation of the Anticoccidial Activity of Sheep Bile against *Eimeria stiedae* Oocysts and Sporozoites of Rabbits: An In Vitro Study"

_vetsci, 2022, doi:10.3390/vetsci9120658_

Round 1

Reviewer 1 Report

Line 21 ‘affect rabbit hepatic’ consider rewording. Liver cells perhaps?

Line 23 italicise Eimeria

Line 44-48 References required

Line 48-49 Unclear if we are still talking about rabbits? Reword.

Line 63 What is TCM? Abbreviation not explained until later.

Line 219 Italicise Eimeria

General Method – you mix between using the subscript for numbers in the chemical formula and not. Please be consistent.

Figure 2 – Spelling error in figure legend. Your sporulation rate for the control is very poor, even less than some of the exposures to Sheep Bile. You do not discuss this later in the discussion or contemplate the reasons why. It seems to have worked better in the later experiments which is good but further discussion is required. You list symbols to denote p values here and yet none of these are present on the figure itself? I can't see which are statistically significant and neither is it well described in the discussion. 

Line 243 – Please add meaning of SB (Sheep bile?) to your figure legend.

Figure 3 and 4 – Spelling errors in sporulation and hours. Same comments regarding figure 2. No symbols to note statistical significance. Does this mean that they are not significant? If so please add this to the legend.

Figure 5 – Improve the title.

Figure 6 Unnecessary to include. Does not provide extra information to Figure 5. Why are we including the number of unsporulated oocysts here? Surely the number of sporulated automatically infers the number unsporulated.

Figure 7 – reword your figure title. Viability and Inhibitory counteract each other. Unsure if this is how viable they are or how inhibited they have been? See above comment regarding displaying significance on the graphs.

Line 280, 288 and 301 italicise Eimeria please check throughout for errors.

Line 289 What do you mean by maximum sporulation inhibition activities? Does this mean no sporozoites were seen? Also more effective than what?

Line 292 – remove the underline and use italics

Author Response

Dear Editor-in-Chief

Dear Reviewer

Thank you for accepting the review in evaluating my manuscript entitled (Evaluation of the anticoccidial activity of the sheep bile liquid against Eimeria Stiedae oocysts and sporozoites of rabbits: in vitro study).

I answered all the questions and mandatory requirements in detail step by step.

Comments and Suggestions for Authors

Reviewer 1:

 All Authors agree with your comments

Line 21 ‘affect rabbit hepatic’ consider rewording. Liver cells perhaps?

Done

Line 23 italicise Eimeria

Done

Line 44-48 References required

Done

Line 48-49 Unclear if we are still talking about rabbits? Reword.

Done

Line 63 What is TCM? Abbreviation not explained until later.

Done

Line 219 Italicise Eimeria

Done

General Method – you mix between using the subscript for numbers in the chemical formula and not. Please be consistent.

Done

Figure 2 – Spelling error in figure legend. Your sporulation rate for the control is very poor, even less than some of the exposures to Sheep Bile. You do not discuss this later in the discussion or contemplate the reasons why. It seems to have worked better in the later experiments which is good but further discussion is required. You list symbols to denote p values here and yet none of these are present on the figure itself? I can't see which are statistically significant and neither is it well described in the discussion.

The data was re-analyzed for oocysts of sporulation and non-sporulation after 48 hours. It was confirmed that the data had been entered correctly. The graph was returned with the insertion of symbols to indicate the value of P value. 

I think that the significant differences between the control and bile are at concentrations of 100% and 75%.

Line 243 – Please add meaning of SB (Sheep bile?) to your figure legend.

Done

Figure 3 and 4 – Spelling errors in sporulation and hours. Same comments regarding figure 2. No symbols to note statistical significance. Does this mean that they are not significant? If so please add this to the legend.

Done

Figure 5 – Improve the title.

Done

Figure 6 Unnecessary to include. Does not provide extra information to Figure 5. Why are we including the number of unsporulated oocysts here? Surely the number of sporulated automatically infers the number unsporulated.

Deleted

Figure 7 – reword your figure title. Viability and Inhibitory counteract each other. Unsure if this is how viable they are or how inhibited they have been? See the above comment regarding displaying significance on the graphs.

Done

Lines 280, 288, and 301 italicize Eimeria please check throughout for errors.

Done

Line 289 What do you mean by maximum sporulation inhibition activities? Does this mean no sporozoites were seen? Also more effective than what?

The sentence has been reformulated

Line 292 – remove the underline and use italics

Done

Reviewer 2 Report

Dears,

Please find enclosed the article with some suggestions and questions.

Best regards,

Author Response

Dear Editor-in-Chief

Dear Reviewer

Thank you for accepting the review in evaluating my manuscript entitled (Evaluation of the anticoccidial activity of the sheep bile liquid against Eimeria Stiedae oocysts and sporozoites of rabbits: in vitro study).

I answered all the questions and mandatory requirements in detail step by step.

Comments and Suggestions for Authors

Reviewer 2:

All Authors agree with your comments

Line 11 repetition

Deleted

Line19 this

Added

Line 22 and 23 italic

Done

Line 44

Done

Line 48

Deleted

 Line 63 Please state the signification of TCM as it is an abbreviation at the first time when you use it and here is the first time. Please consider that for all the abbreviation.

Traditional Chinese Medicine (TCM)

Line 95 please rephrase. replace "work well" by "efficient"

Done

Line 97 per litter of what? water? it's not clear

 Done

Line 105 how much? 1g?

Done

 Line 114 rephrase to avoid "isolated" two time in the same sentence

 Done

Line 122-123: please precise the medium used for flotation

by saturated NaCl solution using

Line 126: not clear...2.5% potassium dichromate (V/V dilution in water?)

When mixing, use 2.5 g(w) of potassium dichromate in 1 ml (v) of distilled water (W/V), and then use the suspension with the bile(v/v) when applying the experiment with oocysts.

Line 134: it is a plate used?

The authors agree with the mistake. We agree with your comments (6-wells plate)

Describe the concentration?

The volume of each well is 5 mL of suspension of the bile sheep diluted by distilled water. We used five concentrations (v/v; 12.5%, 25%, 50%, 75%, and 100%).

Line 157: How? (state the centrifugation characteristics g?)

We used tubes of 15 ml (Falcon) and centrifugation was done at 1008 g for 10 min, several times until the K2Cr2O7 was removed.

Line 160: in g?

1008 g

Line 173: not clear ....is it a normal distribution?

Clarified and have been rephrase.

Reviewer 3 Report

The submitted manuscript presents a study that aims to assess the in vitro potential of healthy sheep bile liquid against sporulation and morphology of Eimeria species, including E. stiedea oocysts. A second aim described in the study attempts to evaluate the in vitro potential of sheep bile liquid against the sporozoites of E. stiedea.

Previous in vitro studies on the effect of animal bile on E. papillata, a coccidian parasite of mice have already been reported. Thus, this study is not novel and could at most provide additional in vitro information on the effect of sheep bile on another coccidian parasite but only at the in vitro level.

The reviewer considers that the submitted manuscript could be greatly improved if an explicit statement of its novelty and originality is written in the paper. Authors could also benefit and improve the manuscript by requesting English editing services or having the manuscript revised by an English-spoken native. Based on this, the results of the study could be of good interest to readers of this journal.  However, there are some points authors are asked to address before the manuscript can be recommended for publication.

 Introduction

 Line 44. It is mentioned that “Coccidiosis causes hepatic Eimeria stiedae in rabbits.” The statement should rather be corrected to “E. stiedae causes hepatic coccidiosis in rabbits.

Line 48. “The mortality rate may go up”. Establish How high? express mortality in % rate.

Line 52. “Eimeria”. Scientific names should be placed in italics. Check this out throughout the whole manuscript, for Eimeria species and other genera/organisms, and include the reference section. For example, see “Eimeria” and/or “E. stidae” in lines 95, 183-184, 189, etc., etc. In addition, use the correct scientific name for the coccidian parasite. In some cases it is written as E. stiedea” (lines 102, 152, 207, 219); “Eimeria pps.” (Line 117); “E. Stiedae” (line 242, 246, 250, 264, 270);E. stalidae” (274); “E. estiedia” (281); “E. stiedia” (Lines 289-290, 345); “E. Stiedae” (line 292); “E. steidia” (line 355).

Line 63. Spell out TCM. First-time use.

Line 68. Is “microbe characteristics” correct? Or should it be anti-microbial characteristics?

Line 90. Is reference 23 correct? The title refers to “Hierarchical fragmentation and differential star formation in the Galactic ‘Snake’: infrared dark cloud G11. 11− 0.12” an Astronomy topic.

Lines 98-99. Check the sentence “In India, however, it has never been tested against rabbit coccidiosis”. This text is not necessary as reference 30 testifies that Toltrazuril has been tested in India.

2. Materials and Methods

2.1. Infrared spectroscopy. This subheading should go after sample (bile) collection. “2.2. Preparation of Bile liquid”.

Line 124. “degrees Celsius”. Can use ‘°C’ symbol

Line 126. “aqueous solution of potassium dichromate containing 2.5%”.  Use “a 2.5% (W/V) potassium dichromate solution”.

Line 132. It is stated that “An in vitro inhibition test of sporulation was utilized to examine the effect of chicken bile…”. The purpose of the study was to test sheep bile. Please clarify, Was it chicken or sheep bile? How was the chicken bile obtained?

Lines 136-137. It is stated that “430 ml of un-sporulated oocysts containing 1x104 oocysts were added to potassium dichromate solution 2.5%”. Please, clarify if 1 x 104 total in 430 ml? That is around 23 oocysts per ml?. Use exponent notation (104).

Line 138. Why is the concentration of sheep bile expressed in W/V instead of V/V, is it not liquid the sheep bile?

Line 141. Use 29ºC instead of 29o C

Lines 144-145. “The number of oocysts left in a total of 50000 oocysts was counted to estimate the percent population of killed oocysts”. The sentence is confusing please rephrase it for clarification. In addition, Clarify the number of oocysts used. In the previous paragraph, it was determined to have only 10,000 un-sporulated oocysts in 430 ml.

Line 154. “As described by you (2014),”. Clarify what reference number the citation refers to.

Line 159. Use 41ºC instead of 41oC

Lines 162-163. Please specify what activities the authors refer to in the statement “to evaluate the in vitro sporozoite activities”.

Line 165. Again: Why is the concentration of sheep bile expressed in W/V instead of V/V, is it not liquid sheep bile?. What concentration was Toltrazuril utilized at?

Line 168. How is the viability of sporozoites determined?, for example, by the Fluorescein Diacetate incorporation technique?

3. Results

Rephrase the first paragraph with something like “...Infrared spectroscopy results of the sheep bile liquid (or Chemical analysis of the sheep bile) showed.....”

Lines 182-183. Specify the name of the strain. Or should it rather be isolate?, stock?. In lines 127-128 it is stated that “In addition, the E. stiedae field isolates were kept alive in the parasitology laboratory by passing them via young rabbits on a regular basis”.

Lines 186-187. The paragraph is confusing: “…which varied according to different concentrations of the tested concentrations. concentration 100% SB. The most effective against E. stiedae was 93% at a concentration of 100% of SB.” Rephrase and confirm that 93% refers to ‘Sporulation inhibition’?

Line 190. “(75%, 50%, and 25 mg/mL)”. Try to be consistent. Use either percent concentration or weight/vol concentration, but not both.

With regard to the results presented in figures 2-6, the reviewer considers they are too many and redundant. One figure showing the graphical view representing the percent inhibition of E.stidae oocyst sporulation for all treatments (sheep bile concentrations) and incubation times, should be sufficient. For example, authors could present only the results of figure 6 but use only the percent inhibition rates of E.stidae oocyst sporulation to better appreciate the effect of SB on the sporulation of E. stidae oocysts!!. On this matter, authors should check spelling in legends to graph axis, and legend to figure. For example, make sure that time is in “hours” and not “houres” and the “Statistical significance asterisks are actually shown in the graph bars”.

Line 219. Rephrase the beginning of the sentence... “The in vitro assay performed to evaluate the effect of SB on the sporozoite viability showed that...”

Lines 224-225. It is stated that “Different concentrations of sheep bile showed concentration-dependent  inhibition for the viability of coccidial sporozoites of different E. stiedea as compared to control groups”. What do authors refer to by “different E. stiedea”?

Lines 230-231. Is stated that “According to our results, most concentrations, including infusion concentrations, exhibited anti-sporozoid al activities against E. stiedae at 1000μg/ml". What do authors mean by “infusion concentrations”?

With regard to the results of the sporozoite viability inhibition assay presented in 3 figures, the reviewer considers they are too many and redundant. Figure 9 should be sufficient to represent the main observations. Change “Viability inhibitory” for Viability inhibition” in the legend of the Y axis and legend of the figure (and throughout the manuscript for that matter). Make sure that time is in “hours” and not “houres” and the “Statistical significance asterisks are actually shown in the graph bars”. The authors should also explain why the negative control showed so high sporozoite viability inhibition percent. (¡¡similar to positive control toltrazuril but much higher than the low sheep bile concentrations tested!!).

Discussion

Lines 281-281. It is stated that “The E. estiedia strain was identified in the examined fecal samples containing oocysts.” Please clarify if this is the case. By definition “Strain refers to a group of organisms within a species that differ in trivial ways from similar groups”. A variation of a particular species that possesses minor differences in its characteristics though still remain distinguishable”. See https://www.biologyonline.com/dictionary

Moreover, in lines 127-128 it is stated that “In addition, the E. stiedae field isolates were kept alive in the parasitology laboratory by passing them via young rabbits on a regular basis”.

Line 285. Check concentration “125, 250, 500, 750, and 1000 g/ml”  use mg instead of g, as 1000 g is a Kilogram!

Line 292. Underlined scientific names are OK, but rather try to be consistent, either underlined or in italics, but not a mixture of both.

Line 330. Correct the term “(nematkodes)”

Lines 335-337. Please check if the paragraph is needed. What is the relationship between essential oils from plants to bile acids in terms of components? Are those components from plants also present in animal bile acids?

Lines 344-345. Similarly, what is the relevance of studies performed with derivates from Curcuma longa when compared to the effect of bile from animal son Eimeria sp?

Conclusion

Could authors elaborate a short paragraph in which they envision how the sheep bile (or bile collected from any other species, for that matter) could be tested in an in vivo  E. steidae rabbit infection setting?

Author Response

Dear Editor-in-Chief

Dear Reviewer

Thank you for accepting the review in evaluating my manuscript entitled (Evaluation of the anticoccidial activity of the sheep bile liquid against Eimeria Stiedae oocysts and sporozoites of rabbits: in vitro study).

I answered all the questions and mandatory requirements in detail step by step.

Comments and Suggestions for Authors

Reviewer 3:

Comments and Suggestions for Authors

All Authors agree with your comments

 Introduction

 Line 44. It is mentioned that “Coccidiosis causes hepatic Eimeria stiedae in rabbits.” The statement should rather be corrected to “E. stiedae causes hepatic coccidiosis in rabbits.

Done

Line 48. “The mortality rate may go up”. Establish How high? express mortality in % rate.

The reference did not mention the deaths % rate

Line 52. “Eimeria”. Scientific names should be placed in italics. Check this out throughout the whole manuscript, for Eimeria species and other genera/organisms, and include the reference section. For example, see “Eimeria” and/or “E. stidae” in lines 95, 183-184, 189, etc., etc. In addition, use the correct scientific name for the coccidian parasite. In some cases it is written as “E. stiedea” (lines 102, 152, 207, 219); “Eimeria pps.” (Line 117); “E. Stiedae” (line 242, 246, 250, 264, 270); “E. stalidae” (274); “E. estiedia” (281); “E. stiedia” (Lines 289-290, 345); “E. Stiedae” (line 292); “E. steidia” (line 355).

The authors agree with the mistake. Done throughout the whole manuscript.

Line 63. Spell out TCM. First-time use.

Done

Line 68. Is “microbe characteristics” correct? Or should it be anti-microbial characteristics?

Edited to anti-microbial characteristics

Line 90. Is reference 23 correct? The title refers to “Hierarchical fragmentation and differential star formation in the Galactic ‘Snake’: infrared dark cloud G11. 11− 0.12” an Astronomy topic.

Correct reference changed

Lines 98-99. Check the sentence “In India, however, it has never been tested against rabbit coccidiosis”. This text is not necessary as reference 30 testifies that Toltrazuril has been tested in India.

Deleted This text

  1. Materials and Methods

2.1. Infrared spectroscopy. This subheading should go after sample (bile) collection. “2.2. Preparation of Bile liquid”.

Done

Line 124. “degrees Celsius”. Can use ‘°C’ symbol

Done

Line 126. “aqueous solution of potassium dichromate containing 2.5%”.  Use “a 2.5% (W/V) potassium dichromate solution”.

When mixing, use 2.5 g(w) of potassium dichromate in 1 ml (v) of distilled water (W/V), and then use the suspension with the bile(v/v) when applying the experiment with oocysts.

Line 132. It is stated that “An in vitro inhibition test of sporulation was utilized to examine the effect of chicken bile…”. The purpose of the study was to test sheep bile. Please clarify, Was it chicken or sheep bile? How was the chicken bile obtained?

Sheep bile

Lines 136-137. It is stated that “430 ml of un-sporulated oocysts containing 1x104 oocysts were added to potassium dichromate solution 2.5%”. Please, clarify if 1 x 104 total in 430 ml? That is around 23 oocysts per ml?. Use exponent notation (104).

430 ml containing 1x104 of un-sporulated oocysts

Line 138. Why is the concentration of sheep bile expressed in W/V instead of V/V, is it not liquid the sheep bile?

Done V/V

Line 141. Use 29ºC instead of 29o C

Done

Lines 144-145. “The number of oocysts left in a total of 50000 oocysts was counted to estimate the percent population of killed oocysts”. The sentence is confusing please rephrase it for clarification. In addition, Clarify the number of oocysts used. In the previous paragraph, it was determined to have only 10,000 un-sporulated oocysts in 430 ml.

10,000 un-sporulated oocysts in 430 ml

Line 154. “As described by You (2014),”. Clarify what reference number the citation refers to.

Done

Line 159. Use 41ºC instead of 41oC

Done

Lines 162-163. Please specify what activities the authors refer to in the statement “to evaluate the in vitro sporozoite activities”.

The phrase has been reformulated and the clarified it

Line 165. Again: Why is the concentration of sheep bile expressed in W/V instead of V/V, is it not liquid sheep bile? What concentration was Toltrazuril utilized at?

V/V , Toltrazuril 5 mg/mL

Line 168. How is the viability of sporozoites determined? for example, by the Fluorescein Diacetate incorporation technique?

The viability of sporozoites is determined by giving sporozoites to rabbits and following them until oocysts descend, thus our inferring the activity and validity of sporozoites before starting the experiment, and then confirming later after obtaining different concentrations results through their experiment on rabbits.

  1. Results

Rephrase the first paragraph with something like “...Infrared spectroscopy results of the sheep bile liquid (or Chemical analysis of the sheep bile) showed.....”

Done

Lines 182-183. Specify the name of the strain. Or should it rather be isolate?, stock?. In lines 127-128 it is stated that “In addition, the E. stiedae field isolates were kept alive in the parasitology laboratory by passing them via young rabbits on a regular basis”.

182-183, The oocysticidal activity of different concentrations of sheep bile against the E. stiedae isolated is summarized in vitro.

127-128, The E. stiedae isolated were kept alive in the parasitology laboratory by passage them via young rabbits on a regular basis.

Lines 186-187. The paragraph is confusing: “…which varied according to different concentrations of the tested concentrations. concentration 100% SB. The most effective against E. stiedae was 93% at a concentration of 100% of SB.” Rephrase and confirm that 93% refers to ‘Sporulation inhibition’?

Done

Line 190. “(75%, 50%, and 25 mg/mL)”. Try to be consistent. Use either percent concentration or weight/vol concentration, but not both.

Done, deleted mg/mL

With regard to the results presented in figures 2-6, the reviewer considers they are too many and redundant. One figure showing the graphical view representing the percent inhibition of E.stidae oocyst sporulation for all treatments (sheep bile concentrations) and incubation times, should be sufficient. For example, authors could present only the results of figure 6 but use only the percent inhibition rates of E.stidae oocyst sporulation to better appreciate the effect of SB on the sporulation of E. stidae oocysts!!. On this matter, authors should check spelling in legends to graph axis, and legend to figure. For example, make sure that time is in “hours” and not “houres” and the “Statistical significance asterisks are actually shown in the graph bars”.

Done

Line 219. Rephrase the beginning of the sentence... “The in vitro assay performed to evaluate the effect of SB on the sporozoite viability showed that...”

Done

Lines 224-225. It is stated that “Different concentrations of sheep bile showed concentration-dependent  inhibition for the viability of coccidial sporozoites of different E. stiedea as compared to control groups”. What do authors refer to by “different E. stiedea”?

Done

Lines 230-231. Is stated that “According to our results, most concentrations, including infusion concentrations, exhibited anti-sporozoid al activities against E. stiedae at 1000μg/ml". What do authors mean by “infusion concentrations”?

Done

With regard to the results of the sporozoite viability inhibition assay presented in 3 figures, the reviewer considers they are too many and redundant. Figure 9 should be sufficient to represent the main observations. Change “Viability inhibitory” for Viability inhibition” in the legend of the Y axis and legend of the figure (and throughout the manuscript for that matter). Make sure that time is in “hours” and not “houres” and the “Statistical significance asterisks are actually shown in the graph bars”. The authors should also explain why the negative control showed so high sporozoite viability inhibition percent. (¡¡similar to positive control toltrazuril but much higher than the low sheep bile concentrations tested!!).

 Done

Discussion

Lines 281-281. It is stated that “The E. estiedia strain was identified in the examined fecal samples containing oocysts.” Please clarify if this is the case. By definition “Strain refers to a group of organisms within a species that differ in trivial ways from similar groups”. A variation of a particular species that possesses minor differences in its characteristics though still remain distinguishable”. See https://www.biologyonline.com/dictionary

Done

Moreover, in lines 127-128 it is stated that “In addition, the E. stiedae field isolates were kept alive in the parasitology laboratory by passing them via young rabbits on a regular basis”.

Done

Line 285. Check concentration “125, 250, 500, 750, and 1000 g/ml”  use mg instead of g, as 1000 g is a Kilogram!

Done

Line 292. Underlined scientific names are OK, but rather try to be consistent, either underlined or in italics, but not a mixture of both.

Done

Line 330. Correct the term “(nematkodes)”

Done

Lines 335-337. Please check if the paragraph is needed. What is the relationship between essential oils from plants to bile acids in terms of components? Are those components from plants also present in animal bile acids?

Done, same of plants components present in bile

Lines 344-345. Similarly, what is the relevance of studies performed with derivates from Curcuma longa when compared to the effect of bile from animal son Eimeria sp?

Done, because using in E. tenella sporozoites

Conclusion

Could authors elaborate a short paragraph in which they envision how the sheep bile (or bile collected from any other species, for that matter) could be tested in an in vivo E. steidae rabbit infection setting?

In vivo: A gavage of rabbits infected with Eimeria stiedae and treated with different concentrations of sheep bile to obtain the best dose can be used as a treatment for rabbits.

Round 2

Reviewer 1 Report

This is a much better draft of this paper and I approve of the changes made. The authors have put in the time and effort to significantly improve this paper. There are still typographical errors within the figures presented. 

1. As per my previous comments check the spelling of 'hours' throughout.

2. Change Non-Sporul to Non-Sporulation throughout.

Further comments

3. Ensure you are consisitent with your spelling of E.stiedae throughout. 

4. I am still concerned about the negative control results. Please explain in your discussion why you believed the negative control showed a high sporozoite count and how this may impact your conclusions.

Author Response

Thank you for an excellent and sharp reviewing to our manuscript. We changed our manuscript according to your suggestions and was edited by one or more of the highly qualified native English-speaking editors at MDPI Please See the certificate.

  1. As per my previous comments check the spelling of 'hours' throughout.

Done

  1. Change Non-Sproul to Non-Sporulation throughout.

Done

Further comments

  1. Ensure you are consistent with your spelling of E.stiedae throughout.

Done throughout “E. stiedae”.

  1. I am still concerned about the negative control results. Please explain in your discussion why you believed the negative control showed a high sporozoite count and how this may impact your conclusions.

Done

Reviewer 3 Report

The submitted manuscript presents a study that aims to assess the in vitro potential of healthy sheep bile liquid against sporulation and morphology of Eimeria species, including E. stiedea oocysts. A second aim described in the study attempts to evaluate the in vitro potential of sheep bile liquid against the sporozoites of E. stiedea.

Previous in vitro studies on the effect of animal bile on E. papillata, a coccidian parasite of mice have already been reported. Thus, this study is not novel and could at most provide additional in vitro information on the effect of sheep bile on another coccidian parasite but only at the in vitro level.

The reviewer considers that the submitted manuscript could be greatly improved if the authors submit the manuscript for English editing services, or have the manuscript revised by an English-spoken native. Based on this, the results of the study could be of interest to readers of this journal.  

While a considerable effort was made by the authors in answering all comments posted by the reviewer, the issue of correctly spelling the scientific names and correct use of italics is still a problem. Check this out throughout the whole manuscript, for Eimeria species and other genera/organisms, including those that appear in the references section. For example, see “E. stiedea” in lines 94, 128, 145, 202; "E. staidae" in line 266; “E. steidae” in line 337; Eimeria spp in line 351, Oryctolagus cuniculus in line 353, etc, etc. Check all reference titles that have a scientific name on them.

Author Response

Thank you for an excellent and sharp reviewing to our manuscript. We changed our manuscript according to your suggestions and was edited by one or more of the highly qualified native English-speaking editors at MDPI Please See the certificate.

The reviewer considers that the submitted manuscript could be greatly improved if the authors submit the manuscript for English editing services, or have the manuscript revised by an English-spoken native. Based on this, the results of the study could be of interest to readers of this journal. 

Done

While a considerable effort was made by the authors in answering all comments posted by the reviewer, the issue of correctly spelling the scientific names and correct use of italics is still a problem. Check this out throughout the whole manuscript, for Eimeria species and other genera/organisms, including those that appear in the references section. For example, see “E. stiedea” in lines 94, 128, 145, 202; "E. stiedea staidae" in line 266; “E. stiedea steidae” in line 337; Eimeria spp in line 351, Oryctolagus cuniculus in line 353, etc, etc. Check all reference titles that have a scientific name on them.

Done
